# Continual Reinforcement Learning with Group Symmetries

Shiqi Liu\*, Mengdi Xu\*, Peide Huang, Xilun Zhang, Yongkang Liu[†], Kentaro Oguchi[†] and Ding Zhao
Department of Mechanical Engineering, Carnegie Mellon University
[†]Toyota Motor North America R&D

*Abstract*—Continual reinforcement learning aims to sequentially learn a variety of tasks, retaining the ability to perform previously encountered tasks while simultaneously developing new policies for novel tasks. However, current continual RL approaches overlook the fact that certain tasks are identical under basic group operations like rotations or translations, especially with visual inputs. They may unnecessarily learn and maintain a new policy for each similar task, leading to poor sample efficiency and weak generalization capability. To address this, we introduce a unique Continual Vision-based Reinforcement Learning method that recognizes Group Symmetries, called COVERS, cultivating a policy for each group of equivalent tasks rather than an individual task. COVERS employs a proximal policy gradient-based (PPO-based) algorithm to train each policy, which contains an equivariant feature extractor and takes inputs with different modalities, including image observations and robot proprioceptive states. It also utilizes an unsupervised task clustering mechanism that relies on 1-Wasserstein distance on the extracted invariant features. We evaluate COVERS on a sequence of table-top manipulation tasks in simulation and on a real robot platform. Our results show that COVERS accurately assigns tasks to their respective groups and significantly outperforms baselines by generalizing to unseen but equivariant tasks in seen task groups. Demos are available on our project page: https://sites.google.com/view/rl-covers/.

## I. INTRODUCTION

Quick adaptation to unseen tasks has been a key objective in the field of reinforcement learning (RL) [11, 19, 18]. RL algorithms are usually trained in simulated environments and then deployed in the real world. However, pre-trained RL agents are likely to encounter new tasks during their deployment due to the nonstationarity of the environment. Blindly reusing policies obtained during training can result in substantial performance drops and even catastrophic failures [42, 16].

Continual RL (CRL), also referred to as lifelong RL, addresses this issue by sequentially learning a series of tasks. It achieves this by generating task-specific policies for the current task, while simultaneously preserving the ability to solve previously encountered tasks [18, 15, 37, 23, 36]. Existing CRL works that rely on the task delineations to handle non-stationary initial states, dynamics or reward functions can greatly boost task performance, particularly when significant task changes occur [37]. However, in realistic task-agnostic settings, these delineations are unknown a prior and have to be identified by the agents. In this work, we explore *how to define and detect task delineations to enhance robots' learning capabilities in task-agnostic CRL.*

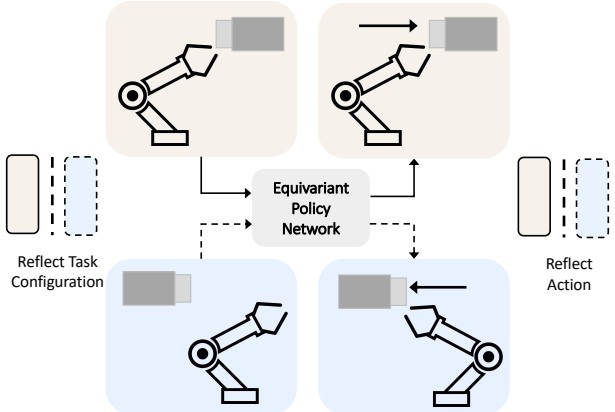

Fig. 1: This example illustrates how group symmetry enhances adaptability. The robot is instructed to close drawers situated in two distinct locations with top-down images as inputs. Considering the symmetry of the drawers' locations around the robot's position, the optimal control policies are equivalent but mirrored.

Our key insight is that robotic control tasks typically preserve certain desirable structures, such as *group symmetries*. Existing CRL approaches typically delineate task boundaries based on statistical measures, such as maximum a posteriori estimates and likelihoods [37, 23]. However, these measures overlook the geometric information inherent in task representations, which naturally emerge in robotic control tasks, as demonstrated in Figure 1. Consider the drawer-closing example: conventional CRL works using image inputs would treat each mirrored configuration as a new task and learn the task from scratch. Yet, we, as humans, understand that the mirrored task configuration can be easily resolved by correspondingly reflecting the actions. Learning the mirrored task from scratch hampers positive task interference and limits the agent's adaptivity. To address this issue, our goal is to exploit the geometric similarity among tasks in the task-agnostic CRL setting to facilitate rapid adaptation to unseen but geometrically equivalent tasks.

In this work, we propose COVERS, a task-agnostic vision-based CRL algorithm with strong sample efficiency and generalization capability by encoding group symmetries in the state and action spaces. We define a *task group* as the set that contains equivalent tasks under the same group operation, such as rotations and reflections. We state our main contributions

as follows:

1) COVERS grows a PPO-based [26] policy with an equivariant feature extractor for each task group, instead of a single task, to solve unseen tasks in seen groups in a zero-shot manner.

2) COVERS utilizes a novel unsupervised task grouping mechanism, which automatically detects group boundaries based on 1-Wasserstein distance in the invariant feature space.

3) In non-stationary table-top manipulation environments, COVERS performs better than baselines in terms of average rewards and success rates. Moreover, we show that (a) the group symmetric information from the equivariant feature extractor promotes the adaptivity by maximizing the positive interference within each group, and (b) the task grouping mechanism recovers the ground truth group indexes, which helps minimize the negative interference among different groups.

## II. RELATED WORK

**Task-Agnostic CRL.** CRL has been a long-standing problem that aims to train RL agents adaptable to non-stationary environments with evolving world models [28, 27, 8, 24, 38, 16, 17, 20, 1, 29]. In task-agnostic CRL where task identifications are unrevealed, existing methods have addressed the problem through a range of techniques. These include hierarchical task modeling with stochastic processes [37, 23], meta-learning [18, 25], online system identification [40], learning a representation from experience [36, 5], and experience replay [24, 7]. Considering that in realistic situations, the new task may not belong to the same task distribution as past tasks, we develop an ensemble model of policy networks capable of handling diverse unseen tasks, rather than relying on a single network to model dynamics or latent representations. Moreover, prior work often depends on data distribution-wise similarity or distances between latent variables, implicitly modeling task relationships. In contrast, we aim to introduce beneficial inductive bias explicitly by developing policy networks with equivariant feature extractors to capture the geometric structures of tasks.

**Symmetries in RL.** There has been a surge of interest in modeling symmetries in components of Markov Decision Processes (MDPs) to improve generalization and efficiency [21, 22, 30, 31, 33, 34, 41, 32, 43, 9, 13, 14]. MDP homomorphic network [30] preserves equivariant under symmetries in the state-action spaces of an MDP by imposing an equivariance constraint on the policy and value network. As a result, it reduces the RL agent's solution space and increases sample efficiency. This single-agent MDP homomorphic network is then extended to the multi-agent domain by factorizing global symmetries into local symmetries [31]. SO(2)-Equivariant RL [33] extends the discrete symmetry group to the group of continuous planar rotations, SO(2), to boost the performance in robotic manipulation tasks. In contrast, we seek to exploit the symmetric properties to improve the generalization capa-

bility of task-agnostic CRL algorithms and handle inputs with multiple modalities.

## III. PRELIMINARY

**Markov decision process.** We consider a Markov decision process (MDP) as a 5-tuple $(\mathcal{S}, \mathcal{A}, T, R, \gamma)$, where $\mathcal{S}$ and $\mathcal{A}$ are the state and action space, respectively. $T : \mathcal{S} \times \mathcal{A} \to \Delta(\mathcal{S})$ is the transition function, $R : \mathcal{S} \times \mathcal{A} \to \mathbb{R}$ is the reward function, and $\gamma$ is the discount factor. We aim to find an optimal policy $\pi_\theta : \mathcal{S} \to \mathcal{A}$ parameterized by $\theta$ that maximizes the expected return $\mathbb{E}_{\tau \sim \pi_\theta} \left[ \sum_{t=0}^{H-1} \gamma^t r(s_t, a_t) \right]$, where $H$ is the episode length.

**Invariance and equivariance.** Let $G$ be a mathematical group. $f : \mathcal{X} \to \mathcal{Y}$ is a mapping function. For a transformation $L_g : \mathcal{X} \to \mathcal{X}$ that satisfies $f(x) = f(L_g[x]), \forall g \in G, x \in \mathcal{X}$, we say $f$ is invariant to $L_g$. Equivariance is closely related to invariance. If we can find another transformation $K_g : \mathcal{Y} \to \mathcal{Y}$ that fulfills $K_g[f(x)] = f(L_g[x]), \forall g \in G, x \in \mathcal{X}$ then we say $f$ is equivariant to transformation $L_g$. It's worth noting that invariance is a special case of equivariance.

**MDP with group symmetries.** In MDPs with symmetries [21, 22, 30], we can identify at least one mathematical group $G$ of a transformation $L_g : \mathcal{S} \to \mathcal{S}$ and a state-dependent action transformation $K_g^s : \mathcal{A} \to \mathcal{A}$, such that $R(s, a) = R\left(L_g[s], K_g^s[a]\right), T(s, a, s') = T\left(L_g[s], K_g^s[a], L_g[s']\right)$ hold for all $g \in G, s, s' \in \mathcal{S}, a \in \mathcal{A}$.

**Equivariant convolutional layer.** Let $G$ be a Euclidean group, with the special orthogonal group and reflection group as subgroups. We use the equivariant convolutional layer developed by Weiler and Cesa [35], where each layer consists of G-steerable kernels $k : \mathbb{R}^2 \to \mathbb{R}^{c_{out} \times c_{in}}$ that satisfies $k(gx) = \rho_{out}(g)k(x)\rho_{in}\left(g^{-1}\right), \forall g \in G, x \in \mathbb{R}^2$. $\rho_{in}$ and $\rho_{out}$ are the types of input vector field $f_{in} : \mathbb{R}^2 \to \mathbb{R}^{c_{in}}$ and output vector field $f_{out} : \mathbb{R}^2 \to \mathbb{R}^{c_{out}}$, respectively.

**Equivariant MLP.** An equivariant multi-layer perceptron (MLP) consists of both equivariant linear layers and equivariant nonlinearities. An equivariant linear layer is a linear function $W$ that maps from one vector space $V_{in}$ with type $\rho_{in}$ to another vector space with type $\rho_{out}$ for a given group $G$. Formally $\forall x \in V_{in}, \forall g \in G : \rho_{out}(g)Wx = W\rho_{in}(g)x$. Here we use the numerical method proposed by Finzi et al. [12] to parameterize MLPs that are equivariant to arbitrary groups.

## IV. METHODOLOGY

### A. Problem Formulation

We focus on continual learning in table-top manipulation environments, where various tasks are sequentially presented. We hypothesize that the streaming tasks can be partitioned into task groups, each containing tasks that share symmetry with one another. We adopt a realistic setting where a new task group may emerge at each episode, the total number of distinct groups remains unknown and the group may arrive in random orders. The primary objective is to devise an online learning algorithm capable of achieving high performance across all tasks with strong data efficiency. We visualize our CRL setting with table-top manipulation environments in Figure 2.

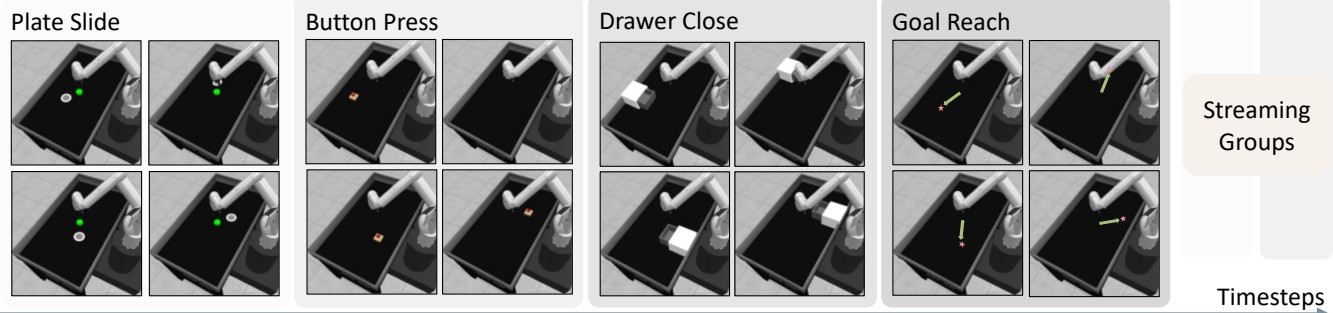

Fig. 2: The continual learning environment setup involves four task groups, including Plate Slide, Button Press, Drawer Close, and Goal Reach. Groups streamingly come in.

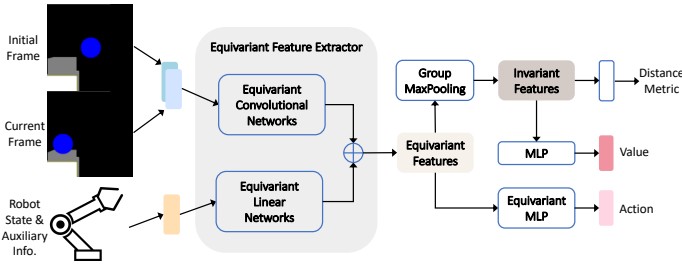

Fig. 3: Equivariant policy network architecture.

## B. Algorithm

We present the pseudocode for COVERS, a task-agnostic continual RL method with group symmetries, in Algorithm 1. COVERS maintains a collection $\Pi = \{(\pi, \mathcal{B})\}$, each element of which comprising a pair of policy $\pi$ and its respective data buffer $\mathcal{B}$. Each policy $\pi$ independently manages one group of tasks, with $\mathcal{B}$ storing the initial frames of the group it oversees. At fixed time intervals, COVERS collects $N_s$ steps in parallel under the current policy $\pi_{cur}$ and stores the first $k$ frames from each episode in the rollout buffer $\mathcal{O}$. Based on $\mathcal{O}$, the algorithm then either (a) creates a new policy for an unseen group and adds it to the collection $\Pi$, or (b) recalls an existing policy from the collection $\Pi$ if the group has been previously encountered. It is worth noting that we assign policies based on initial frames of each episode rather than the full episode rollout. This is because frames corresponding to later timesteps are heavily influenced by the behavior policy and could easily lead to unstable policy assignments. Only maintaining a subset of the rollout trajectories also helps alleviate memory usage.

After the policy assignment, the selected policy $\pi_{cur}$ with parameters $\theta$ is updated based on an online rollout buffer $\mathcal{D}$ and Proximal Policy gradient (PPO) method [26] with loss in Equation 1. $\hat{A}_t$ is the estimated advantage, $\rho_t = \pi_\theta(a_t|s_t)/\pi_{\theta_{old}}(a_t|s_t)$ is the importance ratio and $\epsilon$ is the clip range.

$$\mathcal{L}_{CLIP} = \mathbb{E}_{\tau \sim \mathcal{D}} \Big[ \sum_{t=1}^{H} \min[\rho_t(\theta)\hat{A}_t, \text{clip}(\rho_t(\theta), 1-\epsilon, 1+\epsilon)\hat{A}_t] \Big]. \quad (1)$$

## C. Policy Network Architecture

COVERS utilizes an equivariant policy network that comprises a policy network for predicting actions, a value network approximating values, and an equivariant feature extractor taking multiple modalities. We show the policy architecture in Figure 3 and additional details in Figure 10.

**Equivariant feature extractor.** In real manipulation tasks, the observations typically comprise multiple modalities, such as image observations, robot proprioceptive states, and goal positions represented in vector form. To accommodate these diverse modalities, we designed an equivariant feature extractor $h^{equi}$, that employs an equivariant convolutional network $h^{eConv}$ [35] for image processing, coupled with an equivariant linear network $h^{eMLP}$ [6] to handle vector inputs. The resulting equivariant features from these two pathways are concatenated to form the output of the feature extractor. Formally, $h^{equi}(s) = \text{Concat}(h^{eConv}(s), h^{eMLP}(s))$.

**Invariant value and equivariant policy.** In the context of MDPs involving robotic manipulation tasks with group symmetries, it is known that the optimal value function maintains group invariance, while the optimal policy displays group equivariance [33]. To attain this, both the policy and value networks utilize a shared equivariant feature extractor, designed to distill equivariant features from observations. Subsequently, the value network leverages a group pooling layer to transform these equivariant features into invariant ones, before employing a fully connected layer to generate values. Formally, $h^{inv}(s) = \text{GroupMaxPooling}(h^{equi}(s))$. The policy network, on the other hand, processes the equivariant features with an additional equivariant MLP network to output actions.

## D. Unsupervised Dynamic Policy Assignment

In COVERS, we propose to detect different groups of tasks based on *distances in the invariant feature space*. Such a mechanism facilitates knowledge transfer between tasks in each group. At a fixed episode interval, COVERS selects the policy of the group, whose data buffer $\mathcal{B}$ has the minimal distance in the invariant feature space to the rollout buffer $\mathcal{O}$ collected in the current environment. Note that the invariant features of both $\mathcal{O}$ and $\mathcal{B}$ are obtained through the feature

**Algorithm 1** COVERS: Continual Vision-based RL with Group Symmetries

---

**Input**: Threshold $d_\epsilon$, initial frame number $k$, update interval $N_u$, rollout step size $N_s$

**Output**: collection of policies $\Pi$

**Initialization**: Current policy $\pi_{cur}$ initialized as a random policy with a policy data buffer $\mathcal{B} \leftarrow \varnothing$, policy collection $\Pi \leftarrow \{(\pi_{cur}, \mathcal{B})\}$, number of episodes $n \leftarrow 0$, online rollout buffer $\mathcal{D} \leftarrow \varnothing$

---

1: **while** *task not finish* **do**
2:     $n \leftarrow n + 1$
3:     **if** $n \% N_u = 0$ **then**
4:         Rollout buffer $\mathcal{O} \leftarrow \varnothing$     ▷ Unsupervised Policy Assignment
5:         Rollout $N_s$ steps with $\pi_{cur}$ and get trajectories $\tau = \{(s_0, a_0, \ldots, s_H, a_H)\}$
6:         Append the first $k$ frames of each episode to rollout buffer $\mathcal{O} \leftarrow \{(s_0, \ldots, s_{k-1})\}$
7:         Append the whole episode trajectories $\tau$ to the online rollout buffer $\mathcal{D}$
8:         Calculate the 1-Wasserstein distances $d_i^W(\mathcal{O}, \mathcal{B}_i), \forall \{\pi_i, \mathcal{B}_i\} \in \Pi$ (Equation 2)
9:         Get the minimum distance $d_j^W$ where $j = \arg\min_i d_i^W(\mathcal{O}, \mathcal{B}_i)$
10:         **if** $d_j > d_\epsilon$ **then**
11:             Initialize a new random policy $\pi$ as well as its policy data buffer $\mathcal{B} \leftarrow \mathcal{O}$
12:             $\pi_{cur} \leftarrow \pi$, $\Pi \leftarrow \Pi \cup \{\{\pi, \mathcal{B}\}\}$
13:         **else**
14:             Assign the existing policy and buffer with $\pi_{cur} \leftarrow \pi_j$, $\mathcal{B}_j \leftarrow \mathcal{B}_j \cup \mathcal{O}$
15:             Update $\pi_{cur}$ based on online rollout buffer $\mathcal{D}$ (Equation 1)     ▷ Equivariant Policy Update
16:             $\mathcal{D} \leftarrow \varnothing$
17:     **else**
18:         Sample an episode and append to online rollout buffer $\mathcal{D}$

---

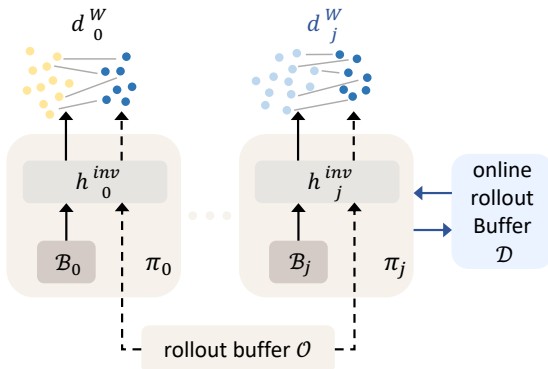

Fig. 4: Calculation of 1-Wasserstein distance and update of selected policy $\pi_j$, whose data has minimal distance to $\mathcal{O}$.

Hence the distance between $\mathcal{O}$ and $\mathcal{B}$ is

$$d^W(\mathcal{O}, \mathcal{B}) = W_1(\mathbf{X}, \mathbf{Y}) = \min_\gamma \langle \gamma, \mathbf{M} \rangle_F$$
$$\text{s.t. } \gamma \mathbf{1} = \mathbf{a}, \gamma^T \mathbf{1} = \mathbf{b}, \gamma \geq 0, \quad (2)$$

where $\mathbf{M}_{i,l} = \|X_i - Y_l\|_2$, $\mathbf{a} = [1/n, \ldots, 1/n]$, $\mathbf{b} = [1/m, \ldots, 1/m]$. $\mathbf{M}$ is the metric cost matrix.

## V. SIMULATION EXPERIMENTS

We validate COVERS's performance in robot manipulation [39] tasks with nonstationary environments containing different objects or following different reward functions. We aim to investigate whether our method can (1) recall stored policy when facing a seen group, as well as automatically initialize a new policy when encountering an unseen group, (2) achieve similar or better performance compared to baselines, and (3) understand the significance of key components of COVERS.

### A. Environment

**Simulation setup.** Our manipulation setup is composed of four groups of tasks. Each group contains four tasks, and all tasks within the same group exhibit rotational or reflectional symmetry with respect to each other. We build environments based on the Meta-World benchmark [39]. Meta-World features a variety of table-top manipulation tasks that require interaction with diverse objects using a Sawyer robot. We show the four groups of tasks in Figure 2 including **Goal Reach** for reaching a goal position, **Button Press** for pressing the button with gripper, **Drawer Close** for closing drawer with gripper, and **Plate Slide** for sliding the plate to a goal position. The goal positions and object locations of tasks in each group are symmetrically arranged around the center of the table.

**States and actions.** The agent receives four kinds of observations: an RGB image captured by a top-down camera centered over the table at each timestep, an RGB image captured by the same camera at the beginning of the episode, the robot state including gripper's 3D coordinates and opening angle, and auxiliary information. The RGB image at the initial step helps alleviate the occlusion problem caused by the movement of the robot. The auxiliary information contains

extractor of $\pi$ as shown in Figure 4. Considering that $\mathcal{O}$ and $\mathcal{B}$ may have a different number of data pairs, we take a probablistic perspective by treating those data buffers as sample-based representations of two distributions and use the Wasserstein distance to measure the distance between those two feature distributions. The invariant features are obtained from the equivariant feature extractor via a group max-pooling operation as shown in Figure 3.

**Wasserstein distance on invariant feature space.** Let $\mathbf{X}$ and $\mathbf{Y}$ be a matrix constructed by invariant features extracted from the state buffer $\mathcal{B}$ of size $n$ and the buffer $\mathcal{O}$ of size $m$. Concretely, $\mathbf{X} = (X_1, X_2, ..., X_n)^{\mathrm{T}}, X_i = h^{inv}(s_i), i \in [n], s_i \in \mathcal{B}$, and $\mathbf{Y} = (Y_1, Y_2, ..., Y_m)^{\mathrm{T}}, Y_l = h^{inv}(s_l), l \in [m], s_l \in \mathcal{O}$. We use the 1-Wasserstein distance [4] to measure the distance between two empirical distributions $\mathbf{X}$ and $\mathbf{Y}$.

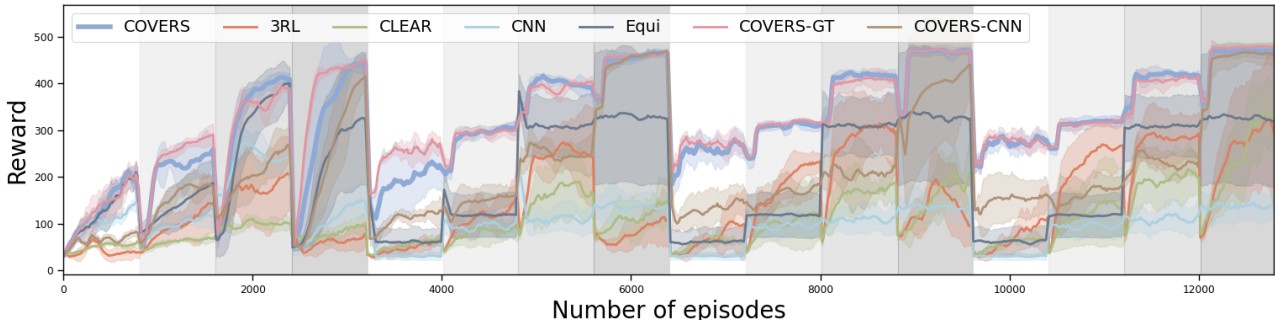

Fig. 5: Training curves for COVERS and other methods. Each background color corresponds to one task group. COVERS shows similar performance with COVERS-GT, which utilizes additional ground truth group indices, and substantially outperforms other baselines.

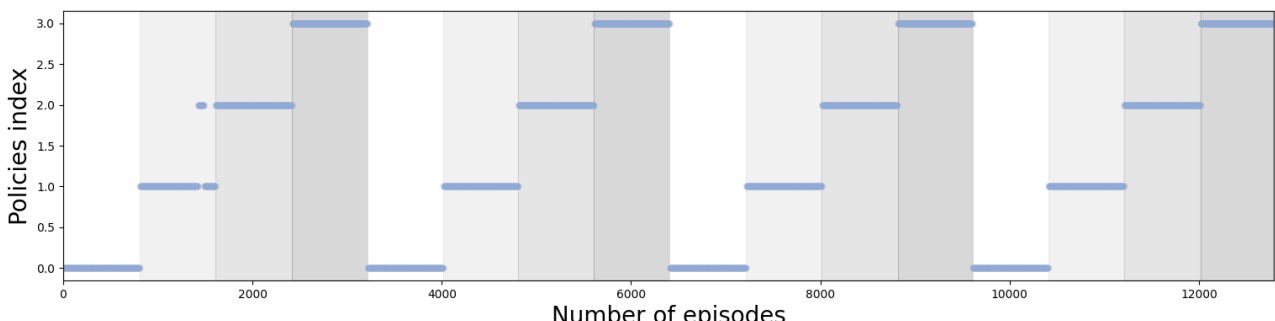

Fig. 6: The selected policies at each episode of COVERS. Each background color corresponds to one task group. The assigned policy indexes remain in alignment with the ground truth ones.

3D goal positions which are only revealed to the agent in Goal Reach since the goal locations are not visualized in the captured image, and are masked out for other groups. To close the sim-to-real gap, we prepossess the RGB images by inpainting robot arms motivated by [2], with details deferred to Section E. A comparison of the original and processed images is visualized in Figure 7. The action is a four-dimensional vector containing the gripper's 3D positions and its opening angle. Considering that we utilize two distinct robots: Sawyer in the simulation and Kinova in the real-world, such an action space and the image preprocessing mechanism help improve transferability between different robot morphologies.

### B. Baselines and Ablations

We compare COVERS with different methods detailed as follows. **3RL** [5], an acronym for Replay-based Recurrent RL, is a state-of-the-art method in CRL with Meta-World tasks that integrates experience replay [24] and recurrent neural networks [3]. Note that we augment **3RL** with a convolutional neural network (CNN) to handle image inputs. In contrast, **CLEAR** [24], a common baseline of CRL, only utilize the experience replay by maintaining a memory buffer to store the experience of the past tasks and oversamples the current tasks to boost the performance in the current one. **Equi** utilizes a single policy with an equivariant feature extractor to solve all tasks. **CNN** utilizes a single policy with a CNN-based feature extractor as a vanilla baseline. We provide the detailed implementation of baselines and hyperparameters in Section D. We compare

with two ablation methods. **COVERS-GT** uses ground truth group labels to assign policies to different groups, which helps ablate the performance of our proposed policy assignment mechanism. **COVERS-CNN** utilizes a vanilla CNN block as the image feature extractor to help ablate the effect of using equivariant feature extractors.

## VI. SIMULATION RESULTS AND ABLATIONS

### A. Results

**Dynamic policy assignments.** Figure 6 shows that when the environment switches to a new group, COVERS quickly detects changes and initializes a new policy for the group. Our method also recalls the corresponding policy from the collection when facing the same group again. Overall, the dynamic policy assignments generated by COVERS align well with the ground truth group labels. However, we observe some instances where the policy assignment does not match the ground truth. This could potentially be attributed to the fact that the feature extractor of each policy may not be able to capture representative features for each group during the early stages of training. Notably, the rate of such misclassifications significantly reduces as the number of training episodes increases.

**Training performance.** We show the training curves of all methods in Figure 5 and the quantitative performance in Table II, including the average success rates and mean rewards. COVERS achieves a much higher episode reward and success rate consistently in different groups than baselines. It is worth

noting that although 3RL performs worse than COVERS, it achieves better performance than baselines with implicit task representations, including Equi, CLEAR, and CNN. This indicates that the explicit task representation used by 3RL, which maps transition pairs to latent variables using an RNN, facilitates the revelation of partial task identifications, thereby enhancing performance. It underscores the significance of task-specific representations in CRL.

In the early stages of training, there isn't a significant performance difference between COVERS and Equi. However, as training progresses, COVERS begins to outperform Equi. This is because COVERS avoids the problem of forgetting through the retraining of policies for each previously encountered task group. A comparison between CNN and Equi reveals that incorporating group symmetries as inductive bias within the equivariant network significantly enhances sample efficiency. This is achieved by only optimizing the policy for the abstracted MDP of each task group.

### B. Ablation Study

**The effect of group symmetric information.** COVERS-CNN devoid of the invariant feature extractor demonstrates lower episodic rewards and success rates when compared with COVERS as shown in Table I and Figure 5. From these results, we conclude that the equivariant feature extractor significantly enhances performance by modeling group symmetry information by introducing beneficial inductive bias through its model architecture.

**The effect of the dynamic policy assignment module** In Figure 5, COVERS's training curve is similar to COVERS-GT, which uses ground truth group indexes as extra prior knowledge. Table I shows that the performance drop due to misclassification is minor considering the small standard deviation and COVERS's performance is within one or two standard deviations of COVERS-GT.

## VII. REAL-WORLD VALIDATION

**Real-world setup.** Our real-world experiment setup utilizes a Kinova GEN3 robotic arm with a Robotiq 2F-85 gripper. The top-down RGB image is captured with an Intel RealSense D345f. Gripper's coordinates and opening angle are obtained through the robot's internal sensors. The real robot setups are demonstrated in Figure 8. We directly deploy the trained policies in simulation to the real world. Table II shows average success rates across 20 trials and shows that our trained policies have strong generalization capability to real-world scenarios. The performance drop compared with simulation experiments may be due to the inconsistent visual features and different scales of robots' action spaces.

## VIII. CONCLUSION

We propose COVERS, a novel Vision-based CRL framework that leverages group symmetries to facilitate generalization to unseen but equivalent tasks under the same group operations. COVERS detects group boundaries in an unsupervised manner based on invariant features and grows policies

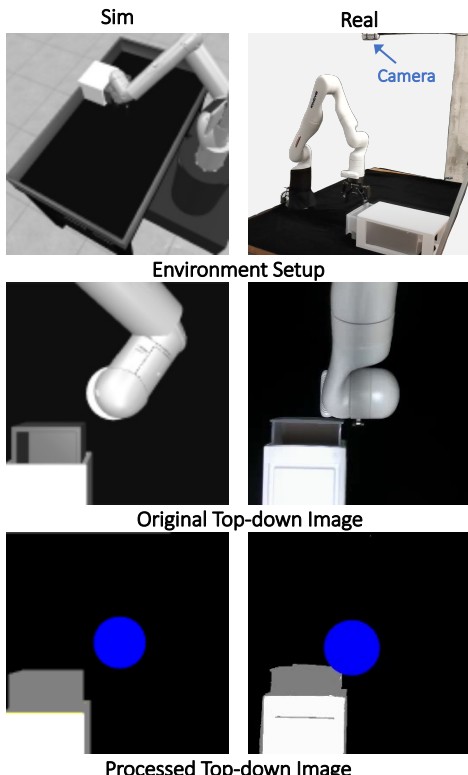

Fig. 7: Image preprocessing to narrow down the sim-to-real gap.

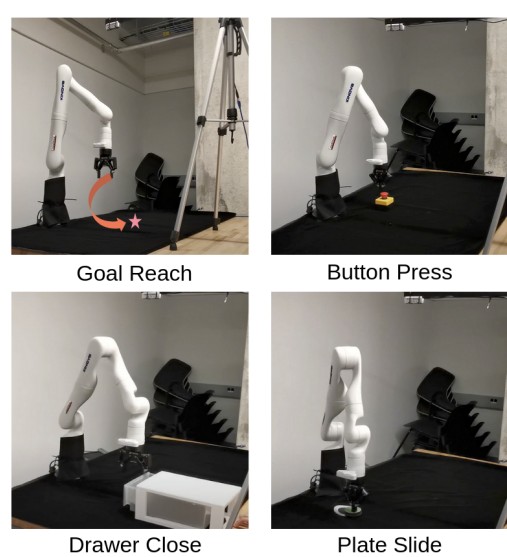

Fig. 8: The real Kinova GEN3 setup with four task groups. The goal point marked in the figure is only disclosed to the agent in Goal Reach as auxiliary information.

for each group of equivalent tasks instead of a single task. We show that COVERS assigns tasks to different groups with high accuracy and has a strong generalization capability, outperforming baselines by a large margin. One limitation of COVERS is that the memory it occupies grows linearly with the number of task groups. However, it is worth noting

TABLE I: Quantitative results showing performances at convergence for different methods.

| Methods | | COVERS | 3RL | CLEAR | CNN | Equi | COVERS-GT | COVERS-CNN |
|---|---|---|---|---|---|---|---|---|
| Plate Slide | Success Rate | **0.97 ± 0.02** | 0.28 ± 0.06 | 0.06 ± 0.03 | 0.03 ± 0.02 | 0.02 ± 0.02 | 0.91 ± 0.03 | 0.62 ± 0.05 |
| | Ave. Reward | **344.04 ± 12.89** | 101.20 ± 7.35 | 65.65 ± 2.23 | 23.44 ± 1.14 | 64.02 ± 5.85 | 337.44 ± 13.87 | 232.25 ± 14.24 |
| Button Press | Success Rate | **0.87 ± 0.04** | 0.52 ± 0.06 | 0.31 ± 0.06 | 0.09 ± 0.03 | 0.01 ± 0.01 | 0.87 ± 0.04 | 0.26 ± 0.05 |
| | Ave. Reward | **323.41 ± 3.48** | 260.80 ± 6.86 | 138.78 ± 12.23 | 91.34 ± 9.34 | 121.13 ± 7.02 | 330.56 ± 2.63 | 181.21 ± 10.83 |
| Drawer Close | Success Rate | 0.82 ± 0.04 | 0.40 ± 0.06 | 0.27 ± 0.05 | 0.16 ± 0.04 | 0.40 ± 0.05 | **0.98 ± 0.02** | 0.56 ± 0.05 |
| | Ave. Reward | 400.09 ± 6.18 | 280.62 ± 6.39 | 216.08 ± 7.68 | 116.33 ± 10.1 | 273.26 ± 9.67 | **417.38 ± 5.6** | 227.3 ± 13.0 |
| Goal Reach | Success Rate | **0.98 ± 0.02** | 0.60 ± 0.06 | 0.58 ± 0.06 | 0.14 ± 0.04 | 0.47 ± 0.05 | 0.97 ± 0.02 | 0.97 ± 0.02 |
| | Ave. Reward | 483.53 ± 1.35 | 322.23 ± 17.33 | 293.5 ± 16.16 | 151.24 ± 14.31 | 306.72 ± 20.34 | **488.02 ± 0.35** | 480.96 ± 1.05 |
| Average | Success Rate | 0.91 ± 0.02 | 0.44 ± 0.03 | 0.30 ± 0.03 | 0.1 ± 0.02 | 0.22 ± 0.02 | **0.93 ± 0.01** | 0.60 ± 0.03 |
| | Ave. Reward | 387.77 ± 5.02 | 241.21 ± 7.39 | 178.5 ± 7.58 | 95.59 ± 5.59 | 191.28 ± 8.23 | **393.35 ± 5.19** | 280.43 ± 8.49 |

TABLE II: Real-world validation results.

| Task Groups | Success Rate |
|---|---|
| Plate Slide | 0.45 ± 0.15 |
| Button Press | 0.60 ± 0.15 |
| Drawer Close | 0.65 ± 0.15 |
| Goal Reach | 0.95 ± 0.07 |

that COVERS still occupies less memory than maintaining a policy buffer for each task by only storing representative data frames such as the initial frames for each task group. Another limitation is that although assuming a top-down camera with a fixed base is widely adopted in existing works, it is hard to fulfill outside of labs. It would be interesting to incorporate more general group operations, such as affine transformation and domain randomization techniques, to handle deformed images. Another interesting future direction is extending our work to continual multi-agent RL settings.

## ACKNOWLEDGMENT

The authors gratefully acknowledge the support from the unrestricted research grant from Toyota Motor North America. The ideas, opinions, and conclusions presented in this paper are solely those of the authors.

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

In this section, we briefly introduce Group and Representation Theory [10] to help understand the policy structure in Section F.

Linear group representations describe abstract groups in terms of linear transformations on some vector spaces. In particular, they can be used to represent group elements as linear transformations (matrices) on that space. A representation of a group $G$ on a vector space $V$ is a group homomorphism from $G$ to $\mathrm{GL}(V)$, the general linear group on V. That is, a representation is a map

$$\rho\colon G \to \mathrm{GL}(V),$$
$$\text{such that} \quad \rho(g_1 g_2) = \rho(g_1)\rho(g_2), \quad \forall g_1, g_2 \in G. \quad (3)$$

Here $V$ is the representation space, and the dimension of $V$ is the dimension of the representation.

### A. Trivial Representation

Trivial representation maps any group element to the identity, i.e.
$$\forall g \in G, \rho(g) = 1. \quad (4)$$

### B. Irreducible Representations

A representation of a group $G$ is said to be irreducible (shorthand as **irrep**) if it has no non-trivial invariant subspaces. For example, given a group $G$ acting on a vector space $V$, $V$ is said to be irreducible if the only subspaces of $V$ preserved under the action of every group element are the zero subspace and $V$ itself. The trivial representation is an irreducible representation and is common to all groups.

### C. Regular Representation

Given a group $G$, the regular representation is a representation over a vector space $V$ which has a basis indexed by the elements of $G$. In other words, if $G$ has $n$ elements (if $G$ is finite), then the regular representation is a representation on a vector space of dimension $n$. An important fact about the regular representation is that it can be decomposed into irreducible representations in a very structured way.

### D. Dihedral Group

The dihedral group $D_n$ is the group of symmetries of a regular n-sided polygon, including $n$ rotations and $n$ reflections. Thus, $D_n$ has $2n$ elements. For example, the dihedral group of a square ($D_4$) includes 4 rotations and 4 reflections, giving 8 transformations in total.

### E. Image Inpainting

To close the sim-to-real gap, we employ a pre-processing technique on camera images, which involves in-painting robotic arms. The process begins by capturing a background image in which the robotic arm is absent from the camera's view. For every time step, a mask that represents the position of each robotic limb is generated, leveraging the 3D locations of individual joints and the projection matrix of the camera. With this mask, we can select all areas devoid of the robotic arm, and subsequently update the background image accordingly. The images are subjected to a color correction process to mitigate any potential color deviations attributable to lighting or reflection. Lastly, a distinct blue circle is overlaid at the gripper's position on the background image to indicate the gripper's location. The entire image in-painting process is shown in Figure 9.

### F. Detailed Policy Architecture

In this section, we present the detailed model architecture including the model sizes and the types of each layer in Figure 10.

In order to make our policy network equivariant under transformations from the finite group $D_2$, we need to choose the appropriate representation for both the network input and output, while also ensuring that the network architecture and operations preserve this equivariance.

The image input is encoded using the trivial representation. The robot state, on the other hand, is encoded with a mixture of different representations: the gripper's position on the z-axis and the gripper's open angle are encoded with the trivial representation since they are invariant to group actions in $D_2$. The gripper's location on the x and y-axes, however, are encoded with two different non-trivial irreducible representations because their values are equivariant to group actions in $D_2$.

The value output is encoded with the trivial representation since the optimal value function should be invariant to group actions [33]. Finally, the action output is encoded with a mixture of different representations. For actions, the gripper movement along the z-axis and the gripper's opening angle are encoded with the trivial representation, while the gripper's location on the x and y-axes are encoded with two different non-trivial irreducible representations, aligning with the input encoding. The distance metric is encoded with trivial representation through the group pooling operation.

### G. Implementation of CLEAR

The CLEAR algorithm [24] addresses the challenge of continual learning by putting data from preceding tasks in a buffer, utilized subsequently for retraining. This method effectively decelerates the rate of forgetting by emulating a continuous learning setting. The specific network architecture for CLEAR is illustrated in Figure 11.

To make CLEAR able to process both images and robot state as input, we introduce a feature extractor, which harmoniously integrates a CNN and an MLP network. This composite feature extractor is carefully designed to contain a similar quantity of learnable parameters to our Equivariant feature extractor.

### H. Implementation of 3RL

The 3RL algorithm [5] can be seen as an improved version of CLEAR, wherein additional historical data is provided to the actor and critic from a dedicated context encoder. This historical data includes $(s_i, a_i, r_i)$, and the context encoder

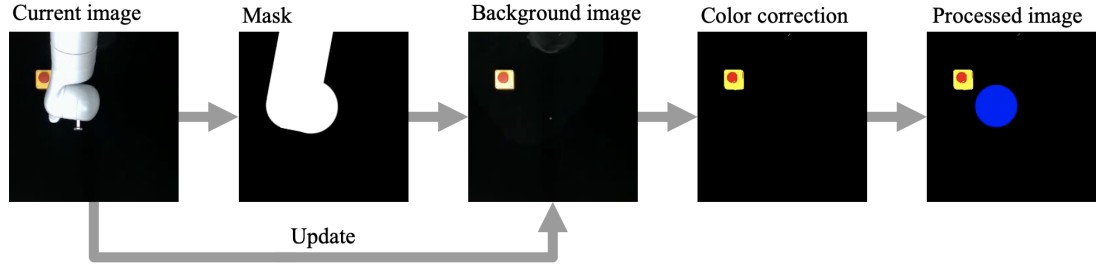

Fig. 9: Image inpainting process.

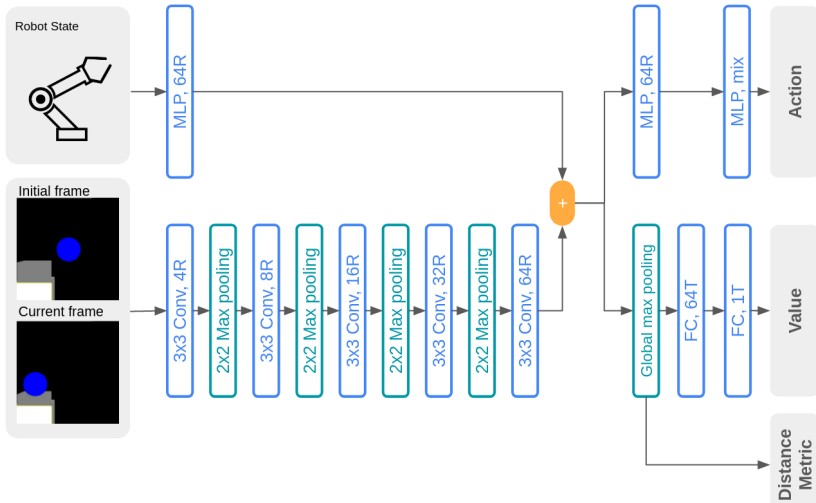

Fig. 10: Detailed equivariant policy network architecture. ReLU nonlinearity is omitted in the figure. A layer with a suffix of R indicates the layer output is in the regular representation. A layer with a suffix of T indicates the layer output is in the trivial representation. A layer with a suffix of 'mix' means the layer output combines different representations.

extracted task specificities from the history data with an RNN network. The specific network architecture for 3RL is illustrated in Figure 12.

*I. Hyperparameters*

We show the hyperparameters of our proposed COVERS in Table III. Moreover, we show the hyperparameters of baselines in Table IV.

TABLE III: COVERS Hyperparameter

| Hyperparameters | Value |
| --- | --- |
| Wasserstein distance threshold $d_\epsilon$ | 1.0 |
| Initial frame number $k$ | 4 |
| Update interval $N_u$ | 1000 |
| Rollout buffer size $N_s$ | 1000 |
| Batch size | 64 |
| Number of epochs | 8 |
| Discount factor | 0.99 |
| Optimizer learning rate | 0.0003 |
| Likelihood ratio clip range $\epsilon$ | 0.2 |
| Advantage estimation $\lambda$ | 0.95 |
| Entropy coefficient | 0.001 |
| Max KL divergence | 0.05 |

TABLE IV: CLEAR and 3RL Hyperparameter

| Hyperparameters | Value |
| --- | --- |
| **Common hyperparameter** | |
| Replay buffer size | 200000 |
| Discount factor | 0.95 |
| Burn in period | 20000 |
| Warm up period | 1000 |
| Batch size | 512 |
| Gradient clipping range | $(-1.0, +1.0)$ |
| Learning rate | 0.0003 |
| Entropy regularization coefficient | 0.005 |
| **3RL Specific Hyperparameters** | |
| RNN's number of layers | 1 |
| RNN's context size | 30 |
| RNN's context length | 5 |

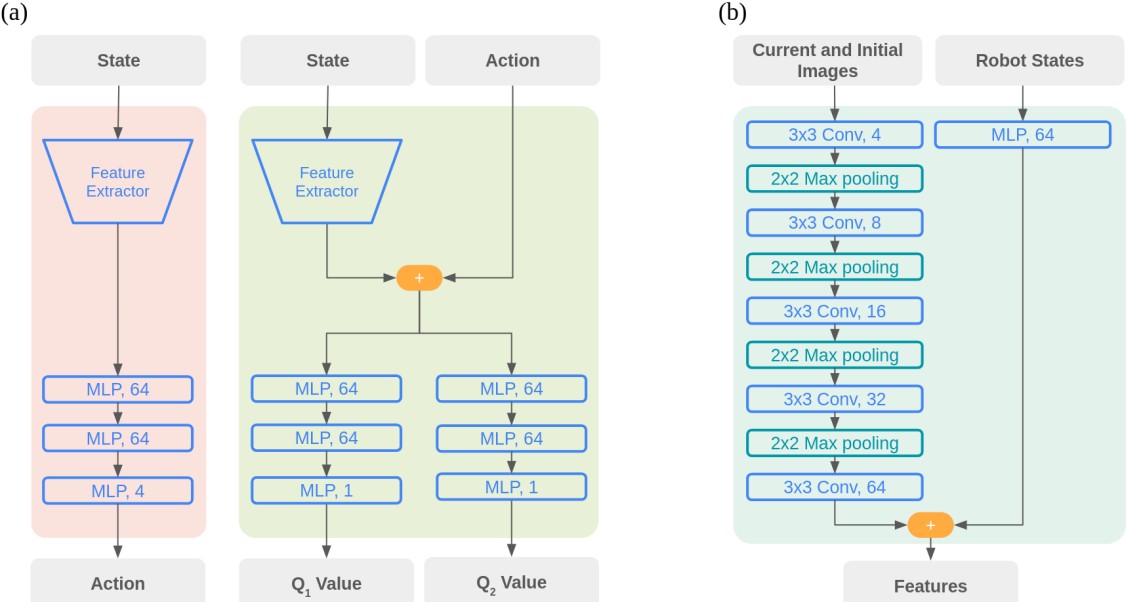

Fig. 11: Network architecture for CLEAR. In (a) we show the network architecture of the actor network and the critic network. In (b) we show the structure of the feature extractor, which consists of both a CNN network and an MLP network. ReLU nonlinearity is omitted in the figure.

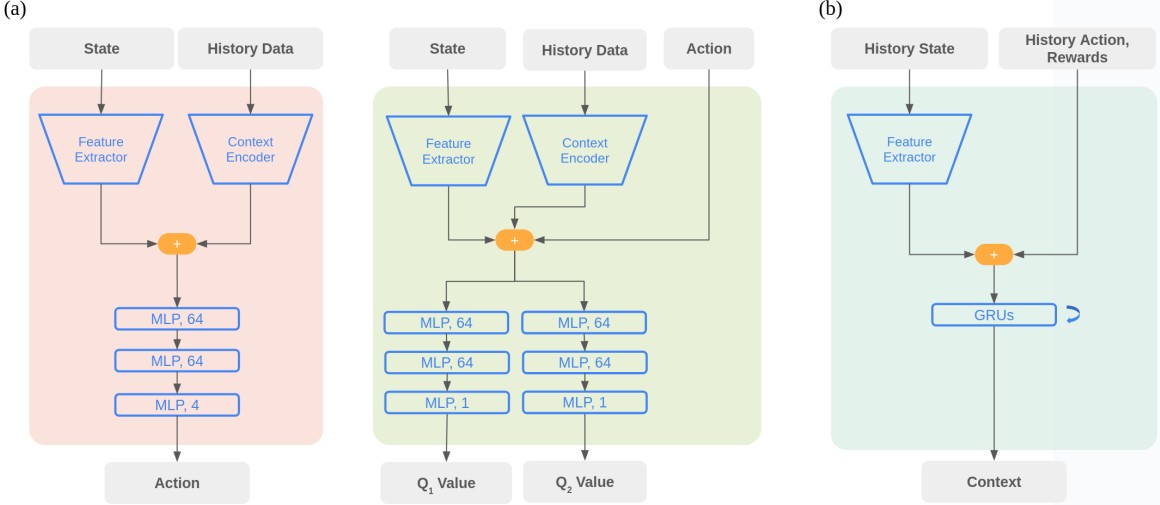

Fig. 12: Network architecture for 3RL. In (a), we illustrate the structure of both the actor and critic networks, whereas (b) highlights the configuration of the context encoder, comprising a feature extractor and GRUs. It's noteworthy that the feature extractor has the same architecture as the CLEAR algorithm, as shown in Figure 11.