# OpenReview forum: "Continual Reinforcement Learning with Group Symmetries"
_roboticsfoundation.org/RSS/2023/Workshop/Symmetry — RSS 2023 Workshop Symmetry_

### Official Review · Reviewer_W1qQ · 2023-06-11
**Review by reviewer W1qQ**

**Rating:** 7
**Confidence:** 5

**Review:**

The paper investigates the application of group symmetries in improving continual reinforcement learning. The paper introduces an equivariant PPO algorithm with an equivariant feature exactor, which is used to group tasks that are invariant under group transformations. Experimental evaluation demonstrates that the proposed method outperforms the baselines in both simulated and real-world scenarios. The paper is well executed and clearly written. Using group symmetries to group tasks that are invariant under group transformation is a very well-motivated and novel idea. The main drawback of the paper is that the novelty of the network architecture shown in Figure 2 compared with the prior work [24] is not very clear. It looks like there are two innovative elements here: 1. there is an extra vector input (potentially represented using a mixed representation as the vector input contains the gripper’s pos and the open angle); 2. the shared encoder (feature exactor) between the critic and the actor. The contribution of the network architecture will be clearer if the authors can clarity such novelties.

I also have some minor comments:
1. The word ‘group’ is used both in ‘group symmetry’ and ‘task group’, which may cause ambiguity at some places.
2. Typo in Section IV. A ‘Observations of these frames are chosen since the are being less impacted …’, ‘the are’ -> ‘they are’.
3. In Fig 2, bottom left, ‘FC, 64’ should be ‘FC, 64T’.
4. In Sec I, the paper says ‘We introduce a novel PPO-based algorithm with a rotation-invariant feature extractor.’ Is the feature exactor only invariant to rotation? Since the paper mentions reflection a couple of times, especially in Section V. B when describing the environments, I suspect that the feature exactor is also invariant to reflection.
5. Building on the previous point, it would be beneficial if the author can clarify which symmetry group is used in the network architecture.

---

### Official Review · Reviewer_n1EY · 2023-06-20
**Leaning Towards Rejection - Insufficient Empirical Validation**

**Rating:** 5
**Confidence:** 4

**Review:**

Summary:
This paper introduces an equivariant PPO architecture and a new unsupervised policy assignment method for continual RL. The authors conduct simulation experiments and its promising results demonstrate the effectiveness of the proposed approach.

Comments:
1. The term "group" is shared by two distinct concepts: symmetry groups and task groups, which can lead to ambiguity for readers. It would be beneficial if the authors could explicitly specify which group they are referring to each time the term is mentioned.

2. The equivariant PPO appears a minor contribution because of its similarity to the prior work presented in [1].

3. The concept of measuring the distance among invariant features rather than within state spaces is interesting because the equivariant network can make the distance measurement to be rotation-invariant. Although the authors give a qualitative explanation of it in VI.B(2), it remains unclear how this approach surpasses previous policy assignment methods such as meta-learning, unsupervised learning, etc. To underscore the efficacy of the proposed method, a comparison between M-Equi and Equi PPO with another policy assignment method would be beneficial.


5. The paper is overall well-written. However, two elements weaken the paper. 1) It is not explicitly stated whether TABLE I and II display simulation or real-world results.  The authors claim that they transfer their model from simulation to the real world without fine-tuning. But the performance drops are not mentioned in the paper, given that the model takes RGB images as input whereas the sim-to-real gap is usually considered to be large in literature. 2) Since the equivariant PPO seems to be an incremental modification of prior work [1], the policy assignment method based on invariant features is considered to be the main contribution. Yet, this part lacks in-depth illustration and comparatively experimental validation.

Conclusion:
This paper demonstrates the potential of utilizing equivariant RL in continual RL, which highly aligns with the theme of this workshop. But the significance of its primary contribution is not sufficiently evident, despite good results are given. The paper's impact could be substantially enhanced by including real-world results and conducting comprehensive ablation studies on invariant-feature-based policy assignment. I incline to reject this paper.

[1] Dian Wang, Robin Walters, and Robert Platt. So (2)-equivariant reinforcement learning. In International Conference on learning representations (ICLR), 2022.

---

### Decision · Program_Chairs · 2023-06-23

**Decision:**

Accept

**Comment:**

Congratulations! We encourage the authors to revise the paper based on the reviewer's feedback.
Your paper will be presented as both a short presentation and a poster. Detailed instructions about the presentation format and camera-ready submission will be sent to you soon.